# Comprehensive intervention for reducing stigma of autism spectrum disorders: Incorporating the experience of simulated autistic perception and social contact

**Masaki Tsujita[1]\*, Miho Homma[2], Shin-ichiro Kumagaya[1], Yukie Nagai[3]**

**1** Research Center for Advanced Science and Technology, The University of Tokyo, Meguro-ku, Tokyo, Japan, **2** LITALICO Laboratory, LITALICO Inc., Meguro-ku, Tokyo, Japan, **3** International Research Center for Neurointelligence, The University of Tokyo, Bunkyo-ku, Tokyo, Japan

\* psy.masaki.tsujita@gmail.com

**Data Availability Statement:** Data are available on Open Science Framework (URL: dx.doi.org/10. 17605/OSF.IO/QZE39).

## Abstract

Stigma of autism spectrum disorders (ASD) is a crucial factor leading to a lower quality of life for individuals with ASD. This research evaluated an intervention effect for the stigma through an experience of simulated autistic visual perception and video-based social contact. The intervention was conducted as an experiential workshop for the ASD simulator. Participants ($N = 217$; 156 women, 61 men; a mean age of 41.3) voluntarily attended the workshop. In the workshop, they received a short lecture on autistic perception, a simulated experience using the ASD simulator, a narrative video of individuals with ASD, and a group discussion. They completed the multidimensional attitudes scale, which was developed to measure four attitudes toward ASD: negative affect, calm, cognitions, and behaviors. The attitudes were measured three times: the period of registration with the workshop, during participation in the workshop, and six weeks after the workshop. With regard to the measure during participation in the workshop, attendees responded to the attitudes at the beginning or end of the workshop, to confirm whether attitudes changes were attributed to the effectiveness of the intervention or artifacts resulting from participation in the workshop. A significant reduction was observed in negative affective attitudes at the end of the workshop and persisted even six weeks afterward, even though not observed in calm, cognitive, and behavioral attitudes. Our findings suggest that implementation of our intervention can bring about reduction of the stigma associated with ASD. Further studies with a random sampling method are needed to validate its generalizability and elaborate the components of the intervention.

## Introduction

Autism spectrum disorders (ASD) are characterized by the Diagnostic and Statistical Manual of Mental Disorders, Fifth Edition (DSM-5) as deficits in the social domain (social

**Funding:** This research was supported by JST CREST "Cognitive Mirroring: Assisting people with developmental disorders by means of self-understanding and social sharing of cognitive processes" (Grant Number: JPMJCR16E2), Japan.

**Competing interests:** The authors have declared that no competing interests exist.

communication and interaction) and non-social domains (repetitive patterns of behavior, interests, or activities, and hyper- or hyporeactivity to sensory input) [1]. It is a considerable problem that individuals with ASD and their caregivers have a lower quality of life than the general population [2–4]. Stigma is a crucial factor leading to a lower quality of life for individuals with ASD, considering previous studies regarding the association of stigma with quality of life in stigmatized social groups [5, 6]. *Stigma* was originally defined as "an attribute that is deeply discrediting" [7]. It was conceptualized by subsequent studies as a relationship between interrelated components of labeling, stereotyping, separating, emotional reactions, status loss, and discrimination in a power situation that allows these processes to unfold [8, 9]. Effective intervention in the stigma of ASD needs to be developed to improve their quality of life.

One of the important factors that relate to stigma of ASD is knowledge about ASD. Previous studies have revealed that lack of knowledge about ASD leads to negative attitudes toward ASD, and intervention in the knowledge brings about decrease of stigma of ASD [10–12]. Recent ASD studies have been focusing upon the non-social domains, especially atypical sensory processing, which was included in DSM-5 as a core deficit. A large number of studies report atypical perception on a variety of modalities, such as audition [13], smell [14], touch [15], taste [16], and vision [17]. A recent literature review suggested that the atypical sensory processing can impact selective attention to social information, which leads to social and communicative difficulties in ASD [18]. Therefore, if individuals with TD acquire knowledge about the atypical sensory processing and its relation to social difficulties in ASD, they will regard sensory-related behaviors and derivative social difficulties as uncontrollable for ASD, and consequently have sympathy responses and helping behavioral intentions [11].

A good solution for individuals with TD to understand autistic perception is to experience a simulated external world perceived by individuals with ASD. A virtual reality system is one of the most appropriate tools for realistically experiencing a simulated external world because it enables us to embody another person's perceptual experiences through interaction between sensory information and voluntary action [19]. Recently, a virtual reality simulator for atypical visual perception in ASD, called the ASD simulator, was developed [20]. In this virtual reality system, visual fields on a head-mounted display are converted to autistic visual fields using visual processing techniques, depending on various features of the primary visual and auditory signals (e.g., brightness, motion, or loudness). The simulated visual fields include various atypical attributes, such as high contrast, high intensity, no color, and blurring, which were referred by interviews with individuals with ASD. An algorithm for adding simulated atypical attributes was developed based on the results of a psychophysical experiment for individuals with ASD [21]. Simulated atypical perception by the ASD simulator holds considerable validity because of substantial community involvement: collaboration of a researcher with ASD as a co-author, identification of atypical visual attributes by interviews with individuals with ASD, and development of algorithm based on the psychophysical experiment for individuals with ASD. Individuals with TD can therefore experience faithfully reproduced autistic visual perception using the ASD simulator, and acquire accurate knowledge about autistic perception.

However, experiencing others' simulated perceptions does not necessarily bring about an improvement in stigma. A study of stigma reduction interventions reported that intervention with simulated hallucinations led to increased positive attitudes or empathy but increased desire for social distancing from individuals with schizophrenia [22]. Interestingly, this previous study also reported that increasing desire for social distancing disappeared if a simulation of hallucinations was combined with empathy manipulation intervention. These results suggest that social contact should be combined with intervention with simulation because it can contextualize atypical experiences provided by simulation [23]. The findings of other studies also support the importance of social contact in an anti-stigma intervention, by both face-to-

face [24] and video-based [25] contact. These studies indicate that social contact can be part of an effective intervention program to reduce the stigma associated with ASD.

The purpose of this research is to evaluate an effective intervention for the stigma of ASD, incorporating two components: experience of simulated autistic visual perception and video-based social contact. Therefore, we held an experiential workshop for the ASD simulator including video-based social contact and measured the attitudes of workshop attendees. If the intervention incorporating the two components is effective in the stigma of ASD, the negative attitudes of workshop attendees toward ASD would decrease after the intervention compared to before the intervention.

## Methods

### Research design

As can be seen in Fig 1, we measured the attitudes three times: the period of registration with the workshop (pretest), during participation in the workshop (workshop period test), and six weeks after the workshop (follow-up test). The pretest was conducted via the Internet when attendees registered for participation in the workshop; thus, it was regarded as a reference to ascertain how the following attitudes would change. The workshop period test was conducted on the workshop day using a paper-and-pencil test. At that time, we randomly assigned the attendees to one of two groups to differentiate when the workshop period test was conducted. One group undertook the workshop test at the beginning of the workshop (beginning group), and the other did so at its end (end group). That is, for the beginning group, both the pretest and workshop period test were taken before the intervention; therefore, the differences between the two tests would be merely measuring time, place, and method (online or paper-and-pencil test).

In contrast, for the end group, the difference between the pretest and workshop period test would reflect the short-term intervention effect because the workshop period test was conducted after the intervention. The purpose of dividing attendees into two groups was to confirm whether attitudes changes on the workshop test were attributed to the short-term intervention effect or artifacts resulting from participation in the workshop, such as nervousness and excitement. As for the follow-up test, both groups conducted it three weeks after the workshop via the Internet; thus, its scores would reflect whether attitudes changes on the workshop tests were transient or lasting. We compared the pretest and follow-up test with data from both groups.

### Participants

The participants comprised 217 individuals with TD who voluntarily attended an experience workshop of the ASD simulator (156 women, 61 men), with a mean age of 41.3 [*SD* = 9.1,

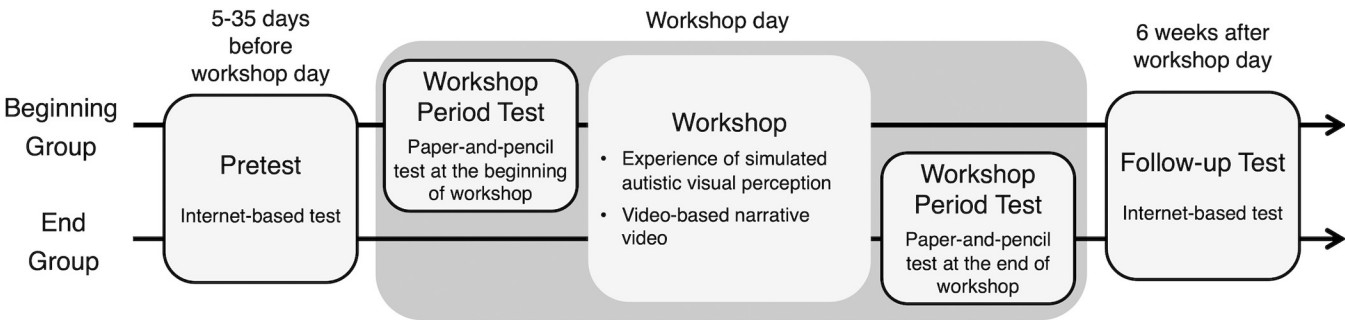

**Fig 1. Schematic illustration of procedure.**

*Min* = 18, *Max* = 65]. They did not paid or get fees for the participation. 88% of the total (192 participants) interacted with individuals with ASD as a family, friend, or co-worker in daily life. Written informed consent was obtained from all participants. This study was approved by the local ethics committee for life science research at the University of Tokyo (approval number: 18–35).

## Materials

**The ASD simulator.** The participants experienced a simulation of autistic visual perception with a head-mounted display (the Samsung Gear VR into which the Samsung Galaxy S8 phone was inserted) and headphones. They watched three videos of daily life scenes: going from a dimly lit entrance room to light outside, waiting at a station platform where a rapid train passed through, and having a meal in a crowded cafeteria. The ASD simulator added various atypical visual attributes to the videos in response to the videos' original visual and auditory signals (Fig 2). The participants watched both original videos and videos with atypical visual attributes.

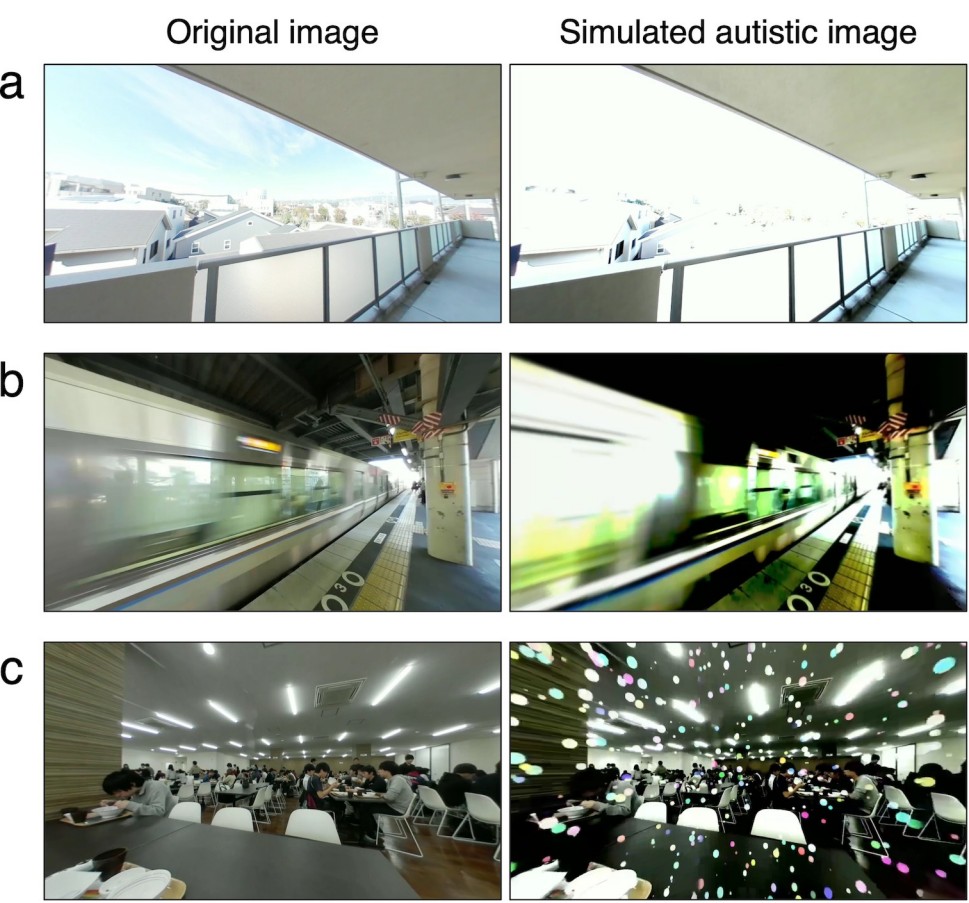

**Fig 2. Images captured from three videos presented with a head-mounted display.** The left column shows original images, and the right column shows simulated autistic images to which the ASD simulator added atypical visual attributes. (a) A scene where a person goes from a room to light outside. High intensity is observed in the simulated autistic image. (b) A scene where a rapid train passes. High contrast and blurring are observed in the simulated autistic image. (c) A scene where a person has a meal in a crowded cafeteria. Visual noises are observed in the simulated autistic image.

**Narrative video of individuals with ASD.**   The participants watched a video in which individuals with ASD narrated their atypical perceptions to have them feel and understand the reality of autistic perception. The video comprises two components. First, individuals with ASD narrated their hypersensitivity while walking around a downtown area and going into a coffee shop with an interviewer. In this part, it was also explained that their hypersensitivity leads to inattention to or unawareness of social information necessary for social interaction and communication. Second, the video showed an introduction of practices to deal with hypersensitivity at a shopping center or a company hiring many individuals with ASD.

**Sociodemographic questionnaire.**   The participants were asked to state their age, gender, and whether they interact with individuals with ASD in their daily lives. Questions about education, financial status, and occupation could not be included in the sociodemographic questionnaire.

**Japanese version of the multidimensional attitudes scale (MAS).**   The MAS was originally developed to measure attitudes toward physical disabilities but was subsequently applied to attitudes toward ASD [26–28]. In this scale, respondents are required to read a vignette describing a social interaction between a protagonist and an individual with a targeting disability or disorder, and then to rate the degree of likelihood that the protagonist might experience multidimensional attitudes in the situation: a dimension of affect (e.g., "Fear" or "Calmness"), cognition (e.g., "He/she seems to be an interesting guy/girl"), and behavior (e.g., "Move to another table"). It has good reliability and validity [26] and has been widely used in Africa [29], Asia [28, 30, 31], and Europe [27, 32–34].

The Japanese version of the MAS [28] presented a vignette describing the interaction between "person A" and an individual with ASD at a coffee shop. Respondents were required to rate the degree of likelihood that "person A" might experience various attitudes in the situation, using a 5-point Likert scale ranging from 1 ("not at all") to 5 ("very much"). It consists of 25 items, which can be subdivided into four attitude subscales (negative affect, calm, cognitions, and behaviors). Note that items in the negative affect and calm subscales belong to the "affect" dimension in the original version of the MAS [26, 28]. Higher scores on all four subscales represent more negative attitudes because items phrased in the opposite direction were reverse-coded. Specifically, a high negative affect subscale score represented more unpleasant emotions, such as "Fear"; a high calm subscale score represented less pleasant emotions, such as "Calmness"; a high cognitions subscale score represented fewer positive thoughts, such as "He/She seems to be an interesting guy/girl"; and a high behaviors subscale score represented more avoidance and escape behaviors, such as "Move to another table."

## Procedure

Participants found recruitment for participation in the workshop through web pages in a developmental disability web portal that a developmental disability support company managed or emails from a company's mailing list. They voluntarily registered for participation 5–35 days before the workshop ($Min$ = 5, $Q_1$ = 24, $Median$ = 26, $Q_3$ = 29, $Max$ = 35). During recruitment, two adjacent days were announced as the workshop days. They were required to decide to participate in either of the two adjacent days. In this registration, they responded to the sociodemographic questionnaire and the MAS as the pretest via the Internet.

On the workshop day, participants first received an explanation of the ASD simulator and a short lecture on autistic perception (45 min). Second, they experienced autistic visual perception using the ASD simulator (60 min). Third, they watched the narrative video of individuals with ASD as a video-based social contact (15 min). Finally, a group discussion was held for 90 min. To sum up, the workshop took 210 min in total.

In the group discussion, the participants were divided into small groups of five to seven participants and one workshop staff member. The workshop staff supported and facilitated active discussions. Participants shared their feelings regarding the simulated experience and the narrative video and their plans for practices to relieve perceptual distress of individuals with ASD in their daily lives.

They responded to the MAS at the beginning or end of the workshop using a paper-and-pencil test as the workshop period test. Two adjacent workshop days were randomly assigned to the beginning or end group. The recruitment for participation in the workshop was conducted three times; thus, the workshop was held for six days in total. The number of participants per day was approximately 30–50. The beginning and end groups comprised 110 and 107 participants, respectively.

Six weeks after participation in the workshop, attendees were required to respond to the MAS via the Internet as a follow-up test. As a result, 85 participants (39.1%) responded to the follow-up test.

## Data analysis

Four sub-scale scores (negative affect, calm, cognitions, and behaviors) were calculated by averaging the item scores of the MAS. A two-way mixed analysis of variance (ANOVA) with two tests (within-subject factor; pretest, workshop period test) and two groups (between-subject factor; beginning group, end group) was conducted for each sub-scale score to test whether and how the attitudes after the workshop increased or decreased from those before the workshop, as compared to simple repetitions of the measurement before the workshop. Furthermore, a paired samples *t*-test with two tests (pretest and follow-up test) was conducted for each sub-scale score to test whether attitudes changed over six weeks after the workshop. A two-way mixed ANOVA with two tests (within-subject factor; pretest, follow-up test) and two groups (between-subject factor; beginning group, end group) was additionally conducted to confirm whether the long-term attitude changes are equivalent between two groups. Significance tests were conducted at an alpha level of 5%. Alpha adjustment for multiple testing to the four sub-scale scores was not applied because each sub-scale score was individually tested [35]. Generalized eta squared statistics ($\eta_g^2$), and Hedges' *g* statistics were estimated as the effect size for the ANOVA and *t*-test, respectively.

## Results

Table 1 shows descriptive statistics of age, gender, and interaction with individuals with ASD for the beginning and end groups. These sociodemographic variables were compared between the two groups to confirm equality. The results showed no significant differences and

**Table 1. Sociodemographic variables of participants (*N* = 217).**

| Sociodemographic variables | Beginning group (*N* = 110) | | End group (*N* = 107) | |
|---|---|---|---|---|
| | *M* / *N* | *SD* / % | *M* / *N* | *SD* / % |
| Age | 41.6 | 8.6 | 41.0 | 10.0 |
| Gender | | | | |
| Female | 77 | 70.0 | 79 | 73.8 |
| Male | 33 | 30.0 | 28 | 26.2 |
| Interaction with individuals with ASD | | | | |
| Yes | 97 | 88.2 | 95 | 88.8 |
| No | 13 | 11.8 | 12 | 11.2 |

**Table 2. Mean and standard deviations of the four subscale scores of three tests.**

|  | Pretest | | Workshop period test | | Follow-up test | |
|---|---|---|---|---|---|---|
|  | *M* | *SD* | *M* | *SD* | *M* | *SD* |
| Beginning group [a] (*N* = 110) |  |  |  |  |  |  |
| Negative affect | 3.67 | 0.64 | 3.59 | 0.52 |  |  |
| Calm | 3.49 | 0.66 | 3.63 | 0.60 |  |  |
| Cognitions | 2.93 | 0.48 | 2.98 | 0.46 |  |  |
| Behaviors | 2.78 | 0.86 | 2.75 | 0.90 |  |  |
| End group [a] (*N* = 107) |  |  |  |  |  |  |
| Negative affect | 3.61 | 0.66 | 3.34 | 0.71 |  |  |
| Calm | 3.40 | 0.62 | 3.49 | 0.61 |  |  |
| Cognitions | 2.95 | 0.60 | 2.96 | 0.47 |  |  |
| Behaviors | 2.60 | 0.87 | 2.50 | 0.85 |  |  |
| Both groups responded to follow-up test [b] (*N* = 85) |  |  |  |  |  |  |
| Negative affect | 3.67 | 0.64 |  |  | 3.52 | 0.55 |
| Calm | 3.47 | 0.63 |  |  | 3.44 | 0.53 |
| Cognitions | 2.93 | 0.52 |  |  | 2.90 | 0.51 |
| Behaviors | 2.62 | 0.87 |  |  | 2.86 | 0.93 |

[a] Participants who did not respond to the follow-up test were included.

[b] Beginning and end groups were merged.

negligible effect sizes for age (Welch's two-sample *t*-test, $t_{(208.96)} = 0.45$, $p = .652$, Hedges' $g = 0.06$), gender ($\chi^2$ test, $\chi^2_{(1)} = 0.39$, $p = .530$, $\varphi = .043$), and interaction with individuals with ASD ($\chi^2$ test, $\chi^2_{(1)} = 0.02$, $p = .889$, $\varphi = -.009$). Therefore, these sociodemographic variables were not included as covariates in the subsequent ANOVA.

Table 2 shows the means and standard deviations of the four subscale scores. "Beginning group" and "End group" in Table 2 enable us to compare changes in sub-scale scores, from the pretest to the workshop period test, between the two groups. These descriptive statistics were calculated using the data in which participants who did not respond to the follow-up test were included. Moreover, "both groups responded to follow-up test" in Table 2, enabling us to compare between the pretest and follow-up test. The beginning and end groups were merged in this comparison because the pretest and follow-up test in the two groups were entirely homogenous (see Fig 1).

## Attitudes changes from pretest to workshop period test

The negative affect score of the workshop period test (*M* = 3.59) was nearly equivalent to that of the pretest (*M* = 3.67) in the beginning group. In contrast, in the end group, the score in the workshop period test (*M* = 3.34) was lower than that of the pretest (*M* = 3.61). A two-way mixed ANOVA revealed a significant main effect for the test ($F_{(1, 215)} = 15.96$, $p < .001$, $\eta_g^2 = .020$) and group ($F_{(1, 215)} = 4.46$, $p = .036$, $\eta_g^2 = .015$) and a significant two-way interaction ($F_{(1, 215)} = 4.58$, $p = .034$, $\eta_g^2 = .006$). The simple effects test for a two-way interaction showed no significant effect for test in the beginning group ($F_{(1, 109)} = 2.39$, $p = .125$, $\eta_g^2 = .005$), while it showed a significant effect for test in the end group ($F_{(1, 106)} = 14.51$, $p < .001$, $\eta_g^2 = .040$). These results indicate that the workshop caused a significant short-term reduction in negative affect, although the effect size for the two-way interaction was small.

In both groups, the calm score of the workshop period test (beginning group: *M* = 3.63, end group: *M* = 3.49) was higher than that of the pretest (beginning group: *M* = 3.49, end group:

$M = 3.40$). A two-way mixed ANOVA showed a significant main effect for test ($F_{(1, 215)} = 5.11$, $p = .025$, $\eta_g^2 = .008$), while no significant main effect for group ($F_{(1, 215)} = 2.87$, $p = .092$, $\eta_g^2 = .009$) nor a significant two-way interaction ($F_{(1, 215)} = 0.21$, $p = .647$, $\eta_g^2 < .001$) was found. These results indicate that participants had already lost calmness, relaxation, or serenity toward individuals with ASD when the workshop started, regardless of participation in the workshop.

The cognitions score of the workshop period test was equivalent to that of the pretest in the end group (pretest: $M = 2.95$, workshop period test: $M = 2.96$) as well as the beginning group (pretest: $M = 2.93$, workshop period test: $M = 2.98$). A two-way mixed ANOVA showed no significant main effect for test ($F_{(1, 215)} = 0.67$, $p = .413$, $\eta_g^2 = .001$) or group ($F_{(1, 215)} < 0.01$, $p = .978$, $\eta_g^2 < .001$) and no significant two-way interaction ($F_{(1, 215)} = 0.34$, $p = .562$, $\eta_g^2 = .001$).

The behaviors score of the workshop period test was equivalent to that of the pretest in the end group (pretest: $M = 2.60$, workshop period test: $M = 2.50$) as well as the beginning group (pretest: $M = 2.78$, workshop period test: $M = 2.75$). A two-way mixed ANOVA revealed a significant main effect for group ($F_{(1, 215)} = 4.74$, $p = .031$, $\eta_g^2 = .015$); however, neither a significant main effect for test ($F_{(1, 215)} = 0.98$, $p = .323$, $\eta_g^2 = .001$) nor a significant two-way interaction ($F_{(1, 215)} = 0.36$, $p = .547$, $\eta_g^2 = .001$) was found.

In summary, short-term stigma reduction by the intervention was observed only in the negative affective attitude. Although reduction from the pretest to the workshop period test was also observed in the calm attitude, it was not due to the intervention. No change was observed in the cognitive and behavioral attitudes.

## Attitudes changes from pretest to follow-up test

The negative affect score at the follow-up test ($M = 3.52$) was still lower than the pretest ($M = 3.67$). The paired samples $t$-test showed a significant difference between the follow-up test and the pretest ($t_{(84)} = 2.28$, $p = .025$, Hedges' $g = 0.25$). These results indicate that the reduced negative affect caused by participating in the workshop persisted even after six weeks. A two-way mixed ANOVA showed a significant main effect for test ($F_{(1, 83)} = 5.12$, $p = .026$, $\eta_g^2 = .016$); while no significant main effect for group ($F_{(1, 83)} = 1.02$, $p = .315$, $\eta_g^2 = .009$) nor a significant two-way interaction ($F_{(1, 83)} = 0.02$, $p = .887$, $\eta_g^2 < .001$) was found.

The calm score of the follow-up test ($M = 3.44$) returned to the pretest level ($M = 3.47$). The paired samples $t$-test showed no significant difference between the two tests ($t_{(84)} = 0.32$, $p = .751$, Hedges' $g = 0.03$), which indicates that the increase in the workshop period test in both groups may be transient and did not persist six weeks after the workshop. A two-way mixed ANOVA revealed a significant main effect for group ($F_{(1, 83)} = 4.09$, $p = .046$, $\eta_g^2 = .025$); however, neither a significant main effect for test ($F_{(1, 83)} = 0.10$, $p = .753$, $\eta_g^2 = .001$) nor a significant two-way interaction ($F_{(1, 83)} < 0.01$, $p = .973$, $\eta_g^2 < .001$) was found.

The cognitions score of the follow-up test ($M = 2.90$) was still equivalent to the pretest ($M = 2.93$). The paired samples $t$-test showed no significant difference between the follow-up and pretest ($t_{(84)} = 0.51$, $p = .610$, Hedges' $g = 0.06$). These results indicate that the three components of the workshop did not affect cognitive attitudes toward individuals with ASD. A two-way mixed ANOVA showed no significant main effect for test ($F_{(1, 83)} = 0.37$, $p = .545$, $\eta_g^2 = .003$) or group ($F_{(1, 83)} = 0.22$, $p = .640$, $\eta_g^2 < .001$) and no significant two-way interaction ($F_{(1, 83)} = 1.58$, $p = .212$, $\eta_g^2 = .006$).

Note that the behaviors score of the follow-up test ($M = 2.86$) was higher than that of the pretest ($M = 2.62$). The paired samples $t$-test found a significant difference in the behaviors score between the pretest and follow-up test ($t_{(84)} = -2.45$, $p = .016$, Hedges's $g = -0.26$). These results suggest that the behavioral attitudes toward individuals with ASD did not change at the

end of the workshop; however, the attitudes increased over six weeks after the workshop. A two-way mixed ANOVA showed a significant main effect for test ($F_{(1, 83)} = 6.20$, $p = .015$, $\eta_g^2 = .018$); while no significant main effect for group ($F_{(1, 83)} = 1.48$, $p = .227$, $\eta_g^2 = .013$) nor a significant two-way interaction ($F_{(1, 83)} = 1.14$, $p = .288$, $\eta_g^2 = .003$) was found.

In summary, persistent stigma reduction was observed only in the negative affective attitude. No change was observed in the calm and cognitive attitudes. Strangely, significant increase over six weeks after the workshop was observed in the behavioral attitude despite no change at the end of the workshop. There was no significant interaction between attitude change and group in all four subscales.

## Discussion

To evaluate an effective intervention for the stigma of ASD, we held a workshop incorporating two components: experience of simulated autistic visual perception and video-based social contact, measuring attendees' attitudes toward individuals with ASD. We assigned the participants to one of two groups (beginning and end groups) to confirm the effectiveness of the intervention. We repeated measuring the attendees' attitudes three times (pretest, workshop period test, and follow-up test) to observe the time-series variation of the attitudes.

Our main finding is that an intervention incorporating the two components effectively reduces negative affective attitudes toward individuals with ASD. The MAS results immediately after the workshop showed a significant reduction in the negative affect from the period of workshop registration compared with the test immediately before the workshop. Moreover, the MAS results six weeks after the workshop showed that the reduction in the negative affect persisted. Although the effect size for the interaction between the test and group factors was small, the effect size for the difference from the pretest was acceptable, both in the test immediately after the workshop and six weeks afterward. These results suggest that participation in the workshop persistently relieves negative affective attitudes toward individuals with ASD.

Unexpectedly, the results of the present study showed an increase in negative attitudes for some dimensions. First, the calm score increased at the workshop period test compared to the pretest. This increase can be interpreted as not a harmful effect of the intervention but an accidental attitude change that was happen regardless of the intervention, by virtue of the special research design in which the participants were assigned to either the beginning or end group. The calm attitude increase at workshop period test was observed in the beginning group as well as the end group. As mentioned in Methods, for the beginning group, both the pretest and workshop period test were taken before intervention. This means that mere arrival at the workshop venue could lead to the calm attitude increase. A possible explanation for this might be that participants became tense and excited throughout the workshop because of their participation and the opportunity to communicate with other participants. They consequently answered with low calming emotions because their scenario was similar to that in the MAS during social interaction with a stranger. Another explanation might be that difference of response method between a paper-and-pencil or online test induced the increase of the calm score. In this study, the pretest and follow-up test was conducted via Internet whereas the workshop period test was conducted using a paper-and-pencil test. Several studies have demonstrated an increase in careless responding in Internet-based research compared to paper-and-pencil research [36–38], therefore it is possible that less careless responding in the workshop period test resulted in the increase of the calm score.

Second, our results showed an increase in the behaviors score only in the follow-up test. This finding is curious because the behaviors score immediately after the workshop did not change, and the other scores did not increase; in fact, the negative affects score decreased at

the follow-up test. This increase might be caused by coincidence between the narrative videos of individuals with ASD and the MAS. The narrative video included content where an individual with ASD cannot listen to a companion's voice in a coffee shop on account of their hypersensitivity to various noises. Likewise, in the MAS, respondents were asked to imagine a situation where the protagonist sits at a table with an individual with ASD in a coffee shop. In each item of the behavior subscale, they answered on the likelihood that avoidant behaviors (e.g., "Move to another table") might arise in the protagonist. Hence, it is possible that participants were aware of various noises in their daily lives for six weeks after the workshop and answered with a high likelihood of avoidant behaviors at the follow-up test in consideration of a stressful situation in which an individual with ASD might interact with the protagonist in a coffee shop. No increase in other attitudes at the follow-up test also supports the possibility that the increase in the behaviors score did not result from rejection of social interaction with an individual with ASD but rather in consideration of their perceptual distress.

Despite limited effectiveness in the negative affective attitude dimension, our results indicate that the intervention with two components is capable of reducing the stigma of ASD. This finding is in line with previous studies that stigma toward mental illness decreased through an intervention with experience of simulated perception [22] or social contact [24, 25]. It is also consistent with a similar study which showed effectiveness of an intervention with experience of simulated perception in decrease of ASD stigma [39]. Moreover, our study demonstrates that the decrease of the stigma persists even six weeks after the intervention. This persistent decrease is comparable to the effectiveness of the intervention with an educational message combined with personalized interaction [40]. It can thus be suggested that experience of simulated perception and social contact are useful as components of an intervention to reduce the stigma of ASD.

## Limitations

There are some limitations to this study that should be addressed in future research. We could not use a random sampling method because we conducted the research in the workshop. Participants recognized workshop announcements via web pages or emails that a developmental disability support company managed, and voluntarily decided to participate in the workshop. Hence, it is likely that most of them were caregivers or families of individuals with ASD and already held positive attitudes toward individuals with ASD. In fact, all attitudes at the pretest in this study were low, except negative affects compared to a previous, Internet-based survey with the Japanese version of the MAS [28]. It means that the sample in this study was not representative of general population, hence our results need to be interpreted with caution.

The results of the comparison between the pretest and the follow-up test must be interpreted with caution because less than half of the participants responded to the test. The reason for such a low response rate is that attendees voluntarily participated in the workshop without pay and thus responded to the follow-up test out of kindness. Therefore, the results of the comparison between the pretest and the follow-up test may not represent the total number of attendees but are biased toward samples of cooperative attendees who were impressed with the workshop.

Our results could not specify which components of the intervention could relieve the stigma of ASD. Although the effectiveness of two components (experience of simulated autistic visual perception and video-based social contact) on stigma has been revealed in the context of various stigma studies (see Introduction), its effectiveness on the stigma of ASD remains unclear. We could not randomly assign participants to different types of intervention because we conducted this study as a workshop to efficiently collect data from unpaid volunteers.

We only measured negative attitudes toward individuals with ASD. In other words, it is unclear whether the measured scores in this study represented negative attitudes toward individuals with ASD or ordinary attitudes toward others in general. Therefore, it is possible that the decrease in the negative affects score in this study does not represent a reduction in negative affective attitudes toward individuals with ASD but merely represents holding positive affective attitudes toward a stranger, regardless of whether the stranger has ASD.

## Future directions and conclusion

Future studies with random sampling from general population should be conducted to confirm generalization of our finding. Future studies with paid volunteers should be conducted to obtain a high response rate to the follow-up test. Future studies examining which components help reduce stigma should be conducted to develop an intervention that is efficient enough to be implemented in various fields. Future studies should measure attitudes toward individuals with TD as well as ASD to compare them.

In conclusion, the present study provided evidence that an intervention incorporating the experience of simulated autistic perception and video-based social contact can relieve negative affective attitudes toward individuals with ASD among individuals with TD. It is worth noting that a single-day workshop brought about persistent relief of the negative affective attitude for six weeks, even though we could not observe relief in the other three attitudes. Our findings suggest that implementing our intervention in education, medical and welfare services, or businesses may bridge knowledge gap about autistic sensory experiences between individuals with ASD and TD. Further studies need to be carried out to validate generalizability of the intervention and elaborate its components.

## Supporting information

**S1 Protocol. A detailed protocol for this research.** dx.doi.org/10.17504/protocols.io. q26g7y9e9gwz/v1.
(DOCX)

**S1 Dataset. Dataset used in the analysis.** dx.doi.org/10.17605/OSF.IO/QZE39.
(DOCX)

## Acknowledgments

We would like to thank Yuhei Suzuki, Jyh-Jong Hsieh, Ayaka Yamamoto, Miya Sabi, and many staff members for helping management and operation of the workshop.

## Author Contributions

**Conceptualization:** Masaki Tsujita, Shin-ichiro Kumagaya.

**Data curation:** Masaki Tsujita, Miho Homma.

**Formal analysis:** Masaki Tsujita.

**Funding acquisition:** Shin-ichiro Kumagaya, Yukie Nagai.

**Investigation:** Masaki Tsujita, Miho Homma.

**Methodology:** Masaki Tsujita, Miho Homma.

**Project administration:** Miho Homma.

**Resources:** Yukie Nagai.

**Software:** Yukie Nagai.

**Supervision:** Shin-ichiro Kumagaya.

**Visualization:** Masaki Tsujita.

**Writing – original draft:** Masaki Tsujita.

**Writing – review & editing:** Shin-ichiro Kumagaya.

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
