## [Decision Letter · Decision Letter 0]

8 Jan 2023

PONE-D-22-20693Comprehensive Intervention for Reducing Stigma of Autism Spectrum Disorders: Incorporating the Experience of Simulated Autistic Perception and Social ContactPLOS ONE

Dear Dr. Tsujita,

Thank you for submitting your manuscript to PLOS ONE. After careful consideration, we feel that it has merit but does not fully meet PLOS ONE’s publication criteria as it currently stands. Therefore, we invite you to submit a revised version of the manuscript that addresses the points raised during the review process. Please submit your revised manuscript by Feb 22 2023 11:59PM. If you will need more time than this to complete your revisions, please reply to this message or contact the journal office at plosone@plos.org. Please include the following items when submitting your revised manuscript:A rebuttal letter that responds to each point raised by the academic editor and reviewer(s). You should upload this letter as a separate file labeled 'Response to Reviewers'.A marked-up copy of your manuscript that highlights changes made to the original version. You should upload this as a separate file labeled 'Revised Manuscript with Track Changes'.An unmarked version of your revised paper without tracked changes. You should upload this as a separate file labeled 'Manuscript'.

We look forward to receiving your revised manuscript.

Kind regards,

Jerritta Selvaraj

Academic Editor

PLOS ONE

and https://journals.plos.org/plosone/s/file?id=ba62/PLOSOne_formatting_sample_title_authors_affiliations.pdf.

“The authors declare that they have no competing interests.”

Reviewers' comments:

Reviewer's Responses to Questions

**Comments to the Author**

1. Is the manuscript technically sound, and do the data support the conclusions?

Reviewer #1: Partly

Reviewer #2: Yes

2. Has the statistical analysis been performed appropriately and rigorously? 

Reviewer #1: No

Reviewer #2: Yes

3. Have the authors made all data underlying the findings in their manuscript fully available?

Reviewer #1: Yes

Reviewer #2: No

4. Is the manuscript presented in an intelligible fashion and written in standard English?

Reviewer #1: Yes

Reviewer #2: No

5. Review Comments to the Author

Reviewer #1: some procedural and statistical errors were found in research paper, the author need special attention to correct these procedural and statistical errors. the method and result section need further clarity and advancement

Reviewer #2: This is an interesting study and the authors have collected a unique dataset using the cutting-edge methodology. The paper is generally well structured. However, in my opinion, the paper has some shortcomings in regard to some text. In the manuscript, numerous text is vague and rambling (e.g. line 53-57, 61-64, etc.) It is suggested to edit the manuscript carefully and rewrite the long sentences to make them more impactful and easy for the reader.

In discussion, it is suggested to cite literature which is supporting your argument even if the hypothesis is not accepted. In the Limitation section, the author has given recommendations. It is suggested to write recommendations under different headings instead of rambling limitations. Furthermore, there is no need of mentioning results in this section. With Given these shortcomings, the manuscript requires revisions.

6. PLOS authors have the option to publish the peer review history of their article (what does this mean?). If published, this will include your full peer review and any attached files.

Reviewer #1: No

Reviewer #2: No

---

## [Author Response · Author response to Decision Letter 0]

17 Mar 2023

>Review Comments to the Author

>Reviewer #1: some procedural and statistical errors were found in research paper, the author need special attention to correct these procedural and statistical errors. the method and result section need further clarity and advancement

We described detailed procedure on protocols.io, and conducted additional analysis to confirm that the extraneous variables are controlled in this study. We also added a new table to display participants’ characteristics.

>Abstract 

>Described the sample selection, procedure, age, and gender base characteristics of the sample briefly. 

We added information of the sample and procedure.

>Introduction is good and well write, need some advance researches.

We replaced some references in Introduction with more recent one.

>Method

>2) Sentence no 134, need rephrasing.

As you mentioned, the first sentence in the section of Research design was unnecessary and helpless to understand the research design. We therefore deleted it.

>3) Explain the standard procedure and protocol, while conducting experimental research.

We described the protocol of this intervention on protocol.io according to recommendation of the editor. We added supporting information section to the manuscript to inform readers about it.

>4) Result discrepancy while assessing the attitude online (pre-test), while the other two-assessment method face to face with individual. Justify? 

The follow-up test was not face to face with individual but was online as well as the pretest. We therefore concluded that decrease of the negative affect score and increase of the behavior score at the follow-up test were not caused by the difference between online or face to face. Rather, it is possible that the difference of assessing method might induce increase of the calm score at the workshop period test. We added the explanation of the possibility at line 159.

>5) Participants demographic characteristics such as age, education, financial status, and occupation are missing. 

Participants’ age was described at line 245 and Table 1. 

We could not include questions about education, financial status, and occupation in the sociodemographic questionnaire for a careless mistake. We added the explanation at line 280.

>6) How the extraneous variables are controlled while conducting experimental method.

As for allocation bias, we randomly assigned two adjacent days to either the beginning or end groups. It seems that there was no allocation bias in this study, considering the results that there was no difference of age, gender, and interaction with individuals with ASD between the beginning and end groups. We also conducted additional analysis to confirm the allocation bias was controlled (see question 8). As for sampling bias, we could not sufficiently control sampling because we recruited participants through a developmental disability support company and as a results most of them were caregivers or families of individuals with ASD. We discussed this sampling bias in the Limitation section starting from line 570.

>7) Need further tabulation on the bases of participant’s characteristics.

We added table of sociodemographic variables as Table 1.

>8) Need further analysis clarify the result?

In order to confirm that the allocation bias did not affect the long-term attitude changes, we additionally conducted a two way mixed ANOVA with two tests (within-subject factor; pretest, follow-up test) and two groups (between-subject factor; beginning group, end group). 

>9) Overall manuscript are not according the standard format APA 7th 

We checked our manuscript and corrected the style and format, according to PLOS ONE submission Guidelines (https://journals.plos.org/plosone/s/submission-guidelines).

>Reviewer #2: This is an interesting study and the authors have collected a unique dataset using the cutting-edge methodology. The paper is generally well structured. However, in my opinion, the paper has some shortcomings in regard to some text. In the manuscript, numerous text is vague and rambling (e.g. line 53-57, 61-64, etc.) It is suggested to edit the manuscript carefully and rewrite the long sentences to make them more impactful and easy for the reader.

>In discussion, it is suggested to cite literature which is supporting your argument even if the hypothesis is not accepted. In the Limitation section, the author has given recommendations. It is suggested to write recommendations under different headings instead of rambling limitations. Furthermore, there is no need of mentioning results in this section. With Given these shortcomings, the manuscript requires revisions.

We agree that the manuscript was vague and rambling, therefore we rewrite it to be clarify and easy to read. In Introduction, we deleted redundant explanations of references and combined some paragraphs. In discussion, we added a paragraph with citations of related literatures to support our argument at line 556-567. We removed recommendations from the Limitations section to the new section named Future Directions and Conclusion. We also deleted mentions of results. Additionally, we deleted a paragraph mentioning internalized stigma in Limitations section because it is extraneous.

---

## [Decision Letter · Decision Letter 1]

2 Jul 2023

Comprehensive Intervention for Reducing Stigma of Autism Spectrum Disorders: Incorporating the Experience of Simulated Autistic Perception and Social Contact

PONE-D-22-20693R1

Dear Dr. Tsujita,

We’re pleased to inform you that your manuscript has been judged scientifically suitable for publication and will be formally accepted for publication once it meets all outstanding technical requirements.

Kind regards,

Filippo Manti

Academic Editor

PLOS ONE

---

## [Editor Report · Acceptance letter]

7 Jul 2023

PONE-D-22-20693R1 

Comprehensive Intervention for Reducing Stigma of Autism Spectrum Disorders: Incorporating the Experience of Simulated Autistic Perception and Social Contact 

Dear Dr. Tsujita:

I'm pleased to inform you that your manuscript has been deemed suitable for publication in PLOS ONE. Congratulations! Your manuscript is now with our production department. 

Kind regards, 

on behalf of

Dr. Filippo Manti 

Academic Editor

PLOS ONE